# Assessment of Heavy Metal Pollution in Suburban River Sediment of Nantong (China) and Preliminary Exploration of Solidification/Stabilization Scheme

Qinqin Xu [1,†], Fengbin Zhao [1,†], Boran Wu [1], Xin Fang [2], Jun Chen [3], Tao Yang [3], Xiaoli Chai [1] and Liqun Yuan [4,*]

1   State Key Laboratory of Pollution Control and Resource Reuse, College of Environmental Science and Engineering, Tongji University, 1239 Siping Road, Shanghai 200092, China; qinqinxu@tongji.edu.cn (Q.X.); fbzhao@tongji.edu.cn (F.Z.); boranwu@tongji.edu.cn (B.W.); xlchai@tongji.edu.cn (X.C.)
2   School of Business, Macau University of Science and Technology, Macau 999078, China; xfang@must.edu.mo
3   Changzhou Drainage Management Office, Changzhou 213017, China; chenjun2022cj@163.com (J.C.); yangtao9004@126.com (T.Y.)
4   Shenzhen Municipal Engineering Corporation, Shenzhen 518033, China
*   Correspondence: yuanliqun@tagen.cn
†   These authors contributed equally to this work.

**Abstract:** Sediments are sinks and sources of pollutants, playing a rather important role in metal migration and transformation. A set of toxic metals of Hg, Pb, Zn, Cr, Cu, Ni and Cd in a suburban river sediment was investigated in the Yangtze River Delta of China, Nantong, and then, the solidification/stabilization scheme and resource-oriented utilization for heavy metal-contaminated sediment were explored. The results showed that all of the metals were apparently higher than the background values. The geo-accumulation index indicated that Ni, Cr, Pb, Cu, Zn and Cd exhibited a none–moderately polluted degree, while Hg corresponded to the moderately contaminated grade. A correlation analysis showed that the contents of metals were not strongly affected by the pH and organic matter content ($p > 0.05$), but they were associated with each other ($p < 0.05$) and might have common natural and anthropogenic sources. Moreover, the leaching experiment revealed that the concentration of Ni exceeded the national standard of China for groundwater, which might cause environmental contamination. Thus, three effective solidification/stabilization formulations for amendments were developed: (1) zero valent iron (9.5% w.w.) and sodium carboxymethylcellulose (0.5% w.w.); (2) sulphate aluminum cement (1% d.w.) and sodium carboxymethylcellulose (0.3% d.w.) and (3) sulphate aluminum cement (1% d.w.), zero valent iron (0.5% d.w.) and sodium carboxymethylcellulose (0.3% d.w.). The findings can provide an effective approach and theoretical basis for the treatment of heavy metal pollution in river sediments.

**Keywords:** sediment; heavy metal; geo-accumulation index; stabilization



## 1. Introduction

With the acceleration of industrialization and urbanization, the concentrations of heavy metals caused by human activities have dramatically increased in recent years [1,2]. Owing to its abundance, persistence and environmental toxicity, the contamination of heavy metals has attracted worldwide attention.

As the largest sinks and sources of heavy metals, sediment is an important medium for metal transformations. In the aquatic environment, heavy metals can be deposited in sediment by adsorption, hydrolysis and coprecipitation processes [3]. In some cases, even more than 99% of heavy metal entering into river can be enriched in sediments [4]. However, heavy metals are not always fixed in the sediment. Their mobility usually depends on various conditions, such as hydrodynamic disturbance, causing sediment re-suspension [5] or bioturbation and chemical factors (temperature, salinity, pH, redox potential, etc.),

generating their desorption from sediment and release into overlying water [6]. In addition, sediments are habitats and food sources for benthic fauna, so that heavy metals may be bioaccumulated and bioamplified through the food chain and end up in the diet of humans [7]. Heavy metals are persistent in river sediment and cannot be degraded by microorganisms, which pose a potential threat to ecological systems and human health [8]. Therefore, it is necessary to investigate the transformation and distribution mechanisms of heavy metals in sediments.

As for soil remediation, ex situ remediation and in situ remediation were adopted for remedying the sediment contaminated by heavy metals. Ex situ remediation can effectively reduce the contents of nutrients, heavy metals and persistent organic matter in sediments [9]. Since the polluted sediment is dredged from the river bed, heavy metal is extracted from or stabilized in the sediment through a series of chemical; physical and biological methods, including washing, electrochemical remediation, flotation, ultrasonic-assisted extraction and immobilization [10]. Though the immobilization methods cannot remove metal from sediment, they are still popularly applied. Common amendments include alkaline agents, phosphate agents, clay minerals, sulfides, heavy metal chelating agent, organic agents, slag, etc. They focus on improving metal stabilization by enhancing metal sorption, precipitation and complexation capacity on sediment. For instance, montmorillonite (MMT) is a good sorbent towards heavy metals via a cation exchange or the formation of inner-sphere complexes through the Si–O and Al–O groups [11]. Lime (LM) affects the adsorption, precipitation and complexation of heavy metals by changing the soil pH, cation exchange capacity, microbial community composition, redox potential and other processes [12]. Zero valent iron (ZVI) metal interactions are based on the corrosion of ZVI by surface complexation, reduction, coprecipitation and cementation [13]. Sodium carboxymethyl-cellulose (CMC) is rich in adsorptive groups such as hydroxyl and carboxyl, and it can also be coordinated with heavy metal ions such as salt materials [14]. Additionally, stabilization/solidification (S/S) involves the addition of cement and cementitious materials (lime, fly ash, blast furnace slag, etc.) for encapsulating the contaminants and improving the engineering properties of sediments [15,16]. The process is associated with cement hydration, cation exchange, flocculation and agglomeration and carbonation [17]. Portland cement has an important hydration product of C-S-H gel that has an extremely high specific surface area adsorption energy and ion exchange capacity, and sulphate aluminum cement (SAC) is hydrated to form large amounts of ettringite that can stabilize particular metallic ions within the ettringite structure [18]. Soil/sediment amendments above (modification or complex formulation) were frequently used in many studies, but the efficient and environmentally friendly heavy metal amendments are still worthy of further exploration.

Nantong (31°41′06″ N–32°42′44″ N, 120°11′47″ E–121°54′33″ E) is one of the important cities in the center of the Yangtze River Delta, covering more than 100 rivers. According to the Nantong Water Resources Bulletin (2019), about 59% of cross-sections of rivers will belong to the water quality standards of grade IV (slight polluted), V (moderate polluted) and inferior V (severe polluted). Among the main seagoing rivers, the water quality of Rutai Canal, Bencha Canal and Beiling River was classified as grade IV, and the Jueqie River was grade V. Thus, the water pollution situation of Nantong is not optimistic. The Public River, as one of the important tributaries, heavy metal pollution in sediment has been rarely reported. Therefore, this study focused on (1) the investigation of the distribution characteristics of heavy metals: (2) assessment of the pollution status of heavy metals and (3) exploration of an effective method for heavy metal stabilization.

## 2. Materials and Methods

### 2.1. Study Area

Rudong County (32°12′ N–32°36′ N, 120°42′ E–121°22′ E) is located in the north wing of the Yangtze River Delta, in the northeast of Nantong City (Jiangsu Province, China). This region belongs to the northern subtropical marine monsoon climate with the annual

average temperature of 15.10 °C. Abundant rainfall averaged more than 1000 mm per year. The landform is a typical coastal plain, spreading in the south of Yellow Sea shore. The river basin is bounded by the Rutai Canal, which runs through the county. To the south of the canal is the Yangtze River Basin that includes Tai Canal, Jiuwei Port, Youwang Port and Jianghai River, accounting for about one thirds of the total basin. To the north of the canal is the Huaihe River Basin that includes Bencha Canal, Nanling River, Yangkou Canal and Juekan River, which accounts for about two thirds of the total basin.

The Public River, as a primary tributary of the Juekan River, is a drainage channel of Rudong suburban area. The contradictions between high-speed development of township enterprises and environmental protection facilities result in quite serious environmental pollution and ecological damage [19]. On the other hand, agricultural non-point sources with the use of chemical fertilizers and pesticides in agricultural production are also the important sources of environmental pollution [19]. Moreover, the disorderly arrangement of houses around the river channel encroaches on the river channel, playing serious impact on flood discharge and waterlogging removal.

### 2.2. Sample Collection

Twice sampling campaigns were taken in the period of April to May in 2020. The sampling stations on Public River are shown in Figure 1. In order to explore the overall pollution status of the Public River, six sampling sites (from S1 to S6) were selected by equidistance to collect surface sediment (box grab method) for the first sampling campaign. Collected sediment was packed in a self-sealing bag and frozen at −20 °C until analysis. Based on the results of this investigation, encrypted sampling sites (from G1 to G8) were selected between S1 and S3 for the second sampling campaign. 8 sediment cores were obtained by column sediment sampler. The length of sampled sediment cores was about 25~30 cm, dividing into two layers, namely superstratum sediment (sr, 0~15 cm) and sub-sediment (sb, >15 cm). In the laboratory, sediment samples were dried at 60 °C and grounded to pass through 200-mesh sieve in an agate mortar.

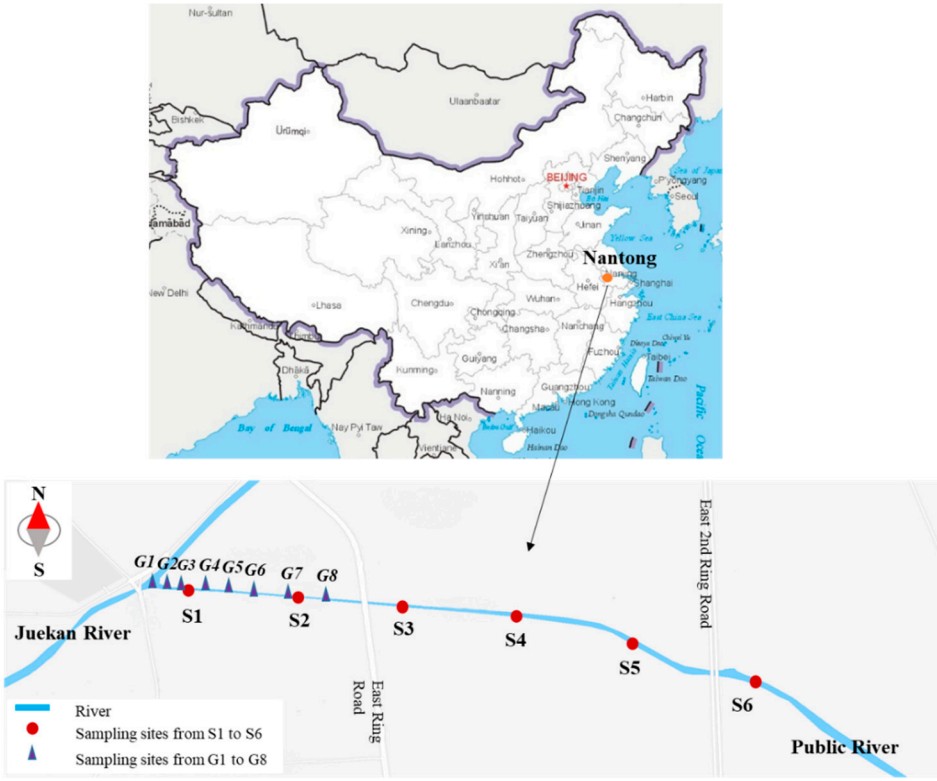

**Figure 1.** Sampling sites for twice sampling campaigns in the Public River.

### 2.3. Solidification/Stabilization Experiment

Two hundred grams of dredged up sediment and the corresponding ratio of amendments were fully stirred and evenly mixed in glass vessels, cured at room temperature for 72 h.

### 2.4. Chemical Analysis

Mercury (Hg) were analyzed by Atomic Fluorescence Spectrometry after microwave dissolution of aqua regia ($HCl:HNO_3$ = 3:1) [20]. Copper (Cu), zinc (Zn), lead (Pb), nickel (Ni) and chromium (Cr) were digested with $HCl–HNO_3–HF$ in microwave digestion system and then detected by flame atomic absorption spectrophotometry [21]. Total cadmium (Cd) was extracted by mixed acid digestion of $HNO_3$, perchloric acid ($HClO_4$), hydrofluoric acid (HF) and HCl, and Graphite Furnace Atomic Absorption Spectrophotometry was used to measure it [22].

The method of HJ/T299-2007 (solid waste-extraction procedure for leaching toxicity-sulphuric acid and nitric acid method) [23] and HJ766-2015 (solid waste-determination of metals-inductively coupled plasma mass spectrometry (ICP-MS)) [24] were adopted to test the leaching behavior of the heavy metals in sediment. The ratio of sediment and leaching agent was 1:10 (L/kg). The pH of leaching agent was $3.20 \pm 0.05$. The solid-liquid mixture was rotated and oscillated for $18 \pm 2$ h with speed of $30 \pm 2$ r/min at the temperature of $23 \pm 2\ ^{\circ}C$. The leachate was filtered and digested with $HNO_3$ and HCl. Then, the supernatant was detected by ICP-MS.

Sediment pH was measured in sediment-water suspension (the ratio of sediment to water was 1:2.5) [25]. Soil moisture was calculated by weight difference method, that dry weight was the samples oven-dried for four hours at 105 °C to constant weight [26]. Soil organic matter (OM) was quantified by oxidation with potassium dichromate in the presence of sulfuric acid, followed by titration with ammonium Fe (II) sulfate [27].

All reagents used in this study were analytical reagent grade. Ultra-pure water (Milli-Q Millipore 18.2 MΩ/cm resistivity) was used for all dilutions. All the plastic and glassware were acid-cleaned ($20\%HNO_3$) for 24 h and rinsed by deionized water for three times. The element standard solutions used for calibration were supplied by Sigma Aldrich.

### 2.5. Quality Assurance and Quality Control (QA/QC)

Accuracy was checked by concurrent analysis of the standard reference materials, repeated test and method blank; the recovery ranged from 94 to 106%. The information on method assurance and analytical facilities are listed in Supplementary Materials Table S1.

### 2.6. Statistical Analysis

The correlations were determined using the simple Pearson correlation coefficient (SPSS software, version 25).

## 3. Results and discussion

### 3.1. Distribution of Heavy Metals in Sediment

Concentrations of heavy metals in public rivers are shown in Table 1. The average concentrations for twice sampling campaigns were $0.737 \pm 0.290$ mg/kg (Hg), $85.8 \pm 23.22$ mg/kg (Pb), $431 \pm 144$ mg/kg (Zn), $303 \pm 166$ mg/kg (Cr), $137 \pm 41.2$ mg/kg (Cu), $149 \pm 81.6$ mg/kg (Ni) and $0.830 \pm 0.500$ mg/kg (Cd), respectively. Levels of heavy metals were apparently higher than the background values (supplementary materials Table S2) in local soil [28]. Heavy metals for Pb, Zn, Cr, Cu, Ni and Cd were 4 to 10 times greater than the background values. Even, the content of Hg was about 30 times the background value. In comparison with other lakes and reservoirs worldwide, the concentrations of Pb, Cr, Cu and Ni were obviously lower than some man-made lakes (Sabalan dam reservoir, Chah Nimeh, Three Gorges Reservoir, etc.) and freshwater lakes (Lake Manzala, Hamahara, East Dongjing Lake, etc.), while the concentration of Zn was far greater than these lakes [29–34].

**Table 1.** Concentrations of heavy metals (*n* = 22) in Public River (mg/kg).

|         | **Hg** | **Pb** | **Zn** | **Cr** | **Cu** | **Ni** | **Cd** |
|---------|--------|--------|--------|--------|--------|--------|--------|
| average | 0.737  | 85.8   | 431    | 303    | 137    | 149    | 0.826  |
| SD      | 0.290  | 23.22  | 144    | 166    | 41.2   | 81.6   | 0.500  |
| min     | 0.095  | 54     | 202    | 138    | 71     | 59     | 0.53   |
| max     | 1.26   | 112    | 708    | 753    | 193    | 373    | 2.18   |

The different concentrations of heavy metals at different sites can be attributed to the changes in terrestrial inputs, hydrodynamic processes and deposition conditions [35]. In the sampling sites from S1 to S6, the highest Hg and Zn levels were observed in S4 (1.26 mg/kg and 413 mg/kg, respectively, Figure 2), and the peak values for Pb, Cr, Cu, Ni and Cd were all shown in S2. In the sampling sites from G1 to G8, the highest concentrations for Zn, Cr and Ni were found at G6, while the lowest levels were recorded at G1. The concentrations of Hg and Pb peaked at G7 and G6, respectively. The maximums of Hg, Pb, Zn, Cr and Ni were about two to three times the minimums. The highest concentration of Cd was 2.18 mg/kg, which was about 4 times as many as the minimal value. Nevertheless, the concentration ranges of Cu (122–193 mg/kg) had small variation. In terms of spatial distribution (Figure 3), almost all heavy metals had no obvious variations between two spatial layers (sr and sb). Particularly, concentrations of Cd at G8, Cr and Ni at G6 differed a lot between sr and sb, which could be attributed to anthropogenic sources [36].

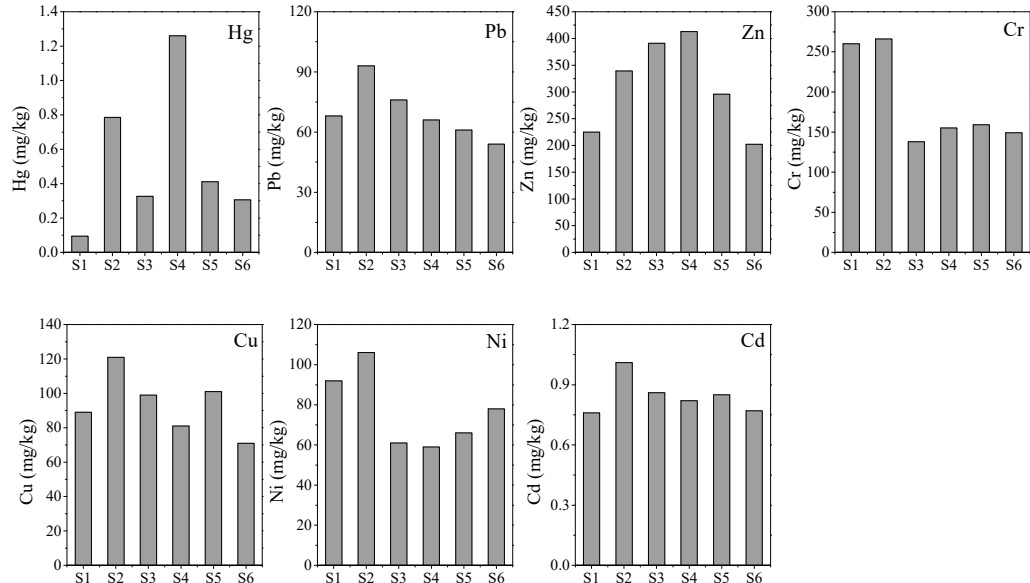

**Figure 2.** Distribution of heavy metals in Public River for the first sampling campaign.

In general, the mobility and availability of heavy metals have been demonstrated to be associated with soil properties, including pH and organic matter content [37]. Sediment in Public River was alkalescent with pH ranging from 7.78 to 8.12. Organic matter (OM) content in sediment ranged 42.10 g/kg to 80.90 g/kg with average of 61.32 ± 12.83 g/kg. Correlation analysis showed that contents of Hg, Pb, Zn, Cr, Cu, Ni and Cd were not strongly affected by the pH and OM contents (*p* > 0.05), implying that the concentration of trace metals in sediment cannot be interpreted simply by changes of these two aspects. Other factors, such as the cation exchange capacity (CEC), oxidation-reduction potential (Eh), the contents of clay minerals, calcium carbonate, Fe and Mn oxides and terrestrial inputs, maybe more important to the distribution of heavy metals [38,39]. However, strong positive correlations were observed for Hg, Pb, Zn, Cr, Cu and Ni (*p* < 0.05), indicating that these metals were associated with each other and may have a common anthropogenic and natural sources [35].

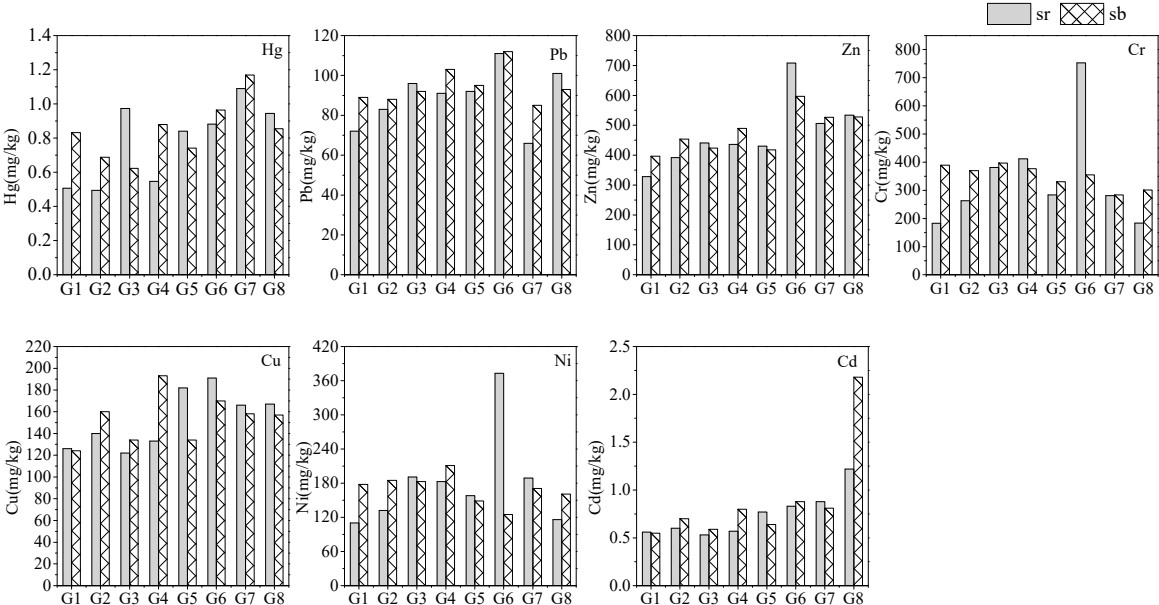

**Figure 3.** Distribution of heavy metals in Public River for the second sampling campaign.

### 3.2. Geo-Accumulation Index ($I_{geo}$)

Geo-accumulation index ($I_{geo}$) is a quantitative index for evaluating the pollution status of heavy metals in water environment sediments, which was calculated using:

$$I_{geo} = \log_2[C_n/(1.5 \times B_n)]$$

where $C_n$ is the measured concentration of the metal in the sediment; $B_n$ is the background or pristine value of the metal, and the value was suggested in Supplementary Materials Table S2 [28]; the constant coefficient of 1.5 was used to assess the natural fluctuations in the content of a given substance in the environment with minimum anthropogenic influence [40]. $I_{geo}$ were classified for seven classes, which were shown in Supplementary Materials Table S3 [40,41].

The $I_{geo}$ results were shown in Figures 4 and 5. The average $I_{geo}$ values for the observed elements were in the increasing order of Cr (0.39) < Pb (0.41) < Ni (0.44) < Cu (0.58) < Zn (0.63) < Cd (0.79) < Hg (1.24) in both sampling campaigns. It implied that Ni, Cr Pb, Cu, Zn and Cd contaminated river sediment mildly, while Hg contaminated the river sediment moderately. All of the $I_{geo}$ values for Ni, Cr, Pb, Cu and Zn, 95% of $I_{geo}$ numbers for Cd, as well as 14% of $I_{geo}$ numbers for Hg ranged from 0 to 1, revealing none-moderately polluted degree. Percentages of $I_{geo}$ values between 1 and 2 for Cd and Hg were 5% and 84%, respectively, corresponding to moderately contaminated grade. The $I_{geo}$ values of Hg were higher than the values (Hg: −0.57) of street deposited sediment in metropolitan region (Shanghai) of China which bears the largest traffic volume, whereas the $I_{geo}$ values of Cr, Cd, Pb, Zn and Cu were obviously lower than it (Cr: 1.11; Cd: 2.23; Pb: 2.32; Zn: 2.44; Cu: 2.48) [40]. As a large industrial city–Porto Alegre, the $I_{geo}$ values of Cr in sediment here even reached up to 5.93 [41], much higher than it in current study. Despite sediment of Sabalan dam reservoir was demonstrated higher metal concentrations, $I_{geo}$ values generally showed no polluted with metals except for Cu (Points A to C, $I_{geo} \cong 0.05$) [29].

### 3.3. Comparison with Relevant Screening Levels

In comparison with critical limits for some foreign countries (Supplementary materials Table S4) [42], 100%, 77.3% and 59.1% of sampling sites for Pb overtopped the critical limits of 50 mg/kg (Eastern Europe, Ireland, Canada, Switzerland, Denmark, Sweden and Finland); 70 mg/kg (Czech Republic) and 85 mg/kg (The Netherlands), respectively. As for Cd, 100%, 45.5% and 13.6% of sampling sites showed a higher concentration than the critical

limits of 0.5 mg/kg (Denmark, Finland, Czech Republic and Canada); 0.8 mg/kg (The Netherlands) and 1 mg/kg (Eastern Europe and Ireland), respectively. Hg concentrations from 86.4% and 45.5% of sampling sites exceeded the critical limit of 0.4 mg/kg in Czech Republic and 0.8 mg/kg in Switzerland. Only 13.6% of the sampling sites for Hg were over the mark of 1 mg/kg in Ireland, whereas all were lower than 2.1 mg/kg in Eastern Europe. However, heavy metals for Zn, Cr, Cu and Ni were all greater than the critical limits for foreign countries in Table S4.

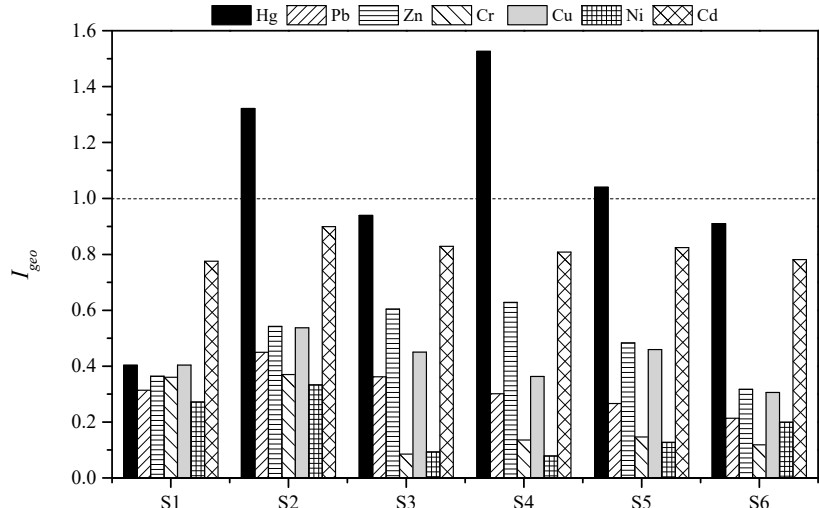

**Figure 4.** $I_{geo}$ class of the metal in the first sampling campaign.

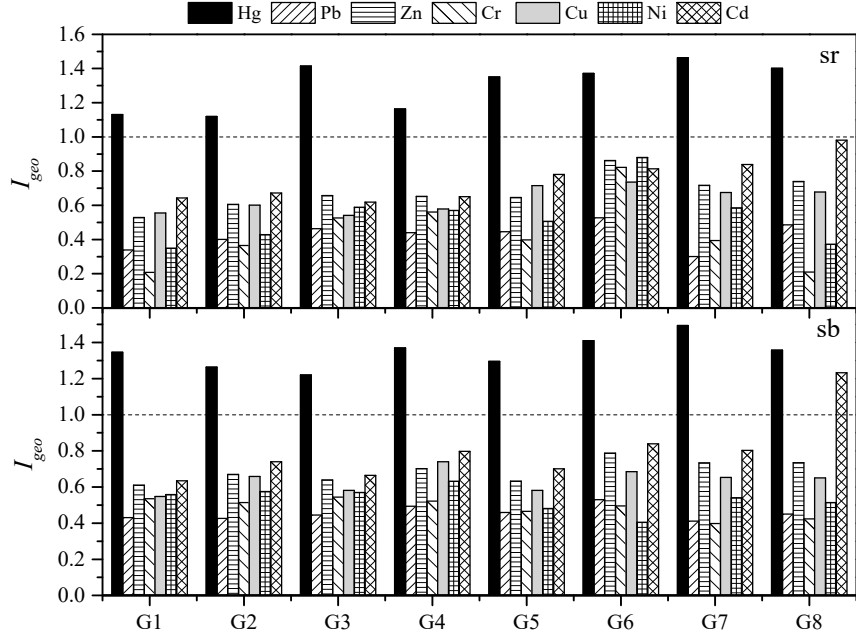

**Figure 5.** $I_{geo}$ class of the metal in the second sampling campaign.

The United States Environmental Protection Agency (USEPA) issued the Ecological Soil Screening Levels (Eco-SSLs) that are the protective concentrations of contaminants for ecological receptors such as plants, soil invertebrates, birds and mammals (Supplementary materials Table S5) [43–48]. More than half of Ni (63.6%) posed a threat to mammalians, but only a few (<10%) overstepped the maximum acceptable toxicant concentration (MATC) value of soil invertebrates and avian. The heavy metals of Pb and Cd were higher than the Eco-SSLs of avian and mammalian, respectively, while both were far less than the Eco-SSLs

of plant and soil invertebrates. All sapling sites for Zn exceeded the Eco-SSLs for four ecological receptors, and so did Cu (95.5%).

The heavy metal levels were evaluated in accordance with national standards of China (Supplementary Materials, Table S6) to seek an appropriate resourced approach for the dredged sediment [49–51]. Cr concentrations in S1 and S2 slightly exceeded the limit value (250 mg/kg) of planting soil for greening (CJ/T 340–2016, Class III, pH > 6.5) (PSG) and Soil Environmental Quality: Risk Control Standard for Soil Contamination of Agricultural Land (GB15618–2018, other pH > 7.5) (SEQA). The Zn concentrations in S2, S3 and S4 were higher than the screening value (300 mg/kg) of *SEQA*, but lower than the limit value (500 mg/kg) of *PSG*. However, the concentrations of Zn in the second sampling campaign (site from G1 to G8) were all over than the screening value (300 mg/kg) of *SEQA*, and even in some sampling sites (G6 to G8), the Zn levels exceeded the limit of value (500 mg/kg) of *PSG*. In almost of all sampling sites (except for G1-sr and G8-sr), the concentrations of Cr went beyond the limit value (250 mg/kg) of *PSG* and *SEQA*. The highest value (753 mg/kg) was indeed about three times more than the threshold (250 mg/kg). Similarly, the contents of Ni were all well above the limit value (150 mg/kg) of *PSG* with the exception of G1-sr, G2-sr, G5-sb, G6-sb and G8-sr, in which three (G3-sr, G4-sb and G6-sr) were higher than the screening value (190 mg/kg) of *SEQA*. Apart from sampling sites G1, G3 and G4-sr, the concentrations of Cd were all over the mark (0.6 mg/kg) of *SEQA*, and only G8 overtopped the limit value (1.2 mg/kg) of *PSG*. However, all of the heavy metals in study area were lower than the threshold of the soil environmental quality: Risk control standard for soil contamination of development land (GB36600-2018; screening value; type II land) (SEQD). Thus, compared with green soil and agriculture soil, development land was a better option for dredged sediment in the Public River to recycling application.

### 3.4. Leaching Behavior of Heavy Metals

The leaching concentrations of metals that exceeded the threshold of *PSG* and *SEQA* (Section 3.3 were analyzed, and the results are shown in Table 2. National standards of China for groundwater and surface water (Supplementary Materials, Table S7) were used to assess the risk of leaching behavior in sediment [51,52]. The leaching concentrations of Cr (6+) in sites from S1 to S2 and from G1 to G8 and Zn in the sites from G6 to G8, as well as Cd in site G8, were below the limit value of the Standard for Groundwater Quality (GB/T 14848-2017; grade IV) (SGQ) and Standard for surface water quality (GB 3838-2002; grade IV) (SSQ). However, leaching concentrations of Ni in G2 (125 µg/L) and G6 (120 µg/L) have already surpassed the threshold (100 µg/L) of *SGQ*, posing a serious threat to water environment. Thus, the dredged sediment shall be solidified/stabilized for construction use.

**Table 2.** Leaching concentration (µg/L) of metals in sediment.

| Species | S1 | G1 | | G2 | | G3 | | G4 | |
|---|---|---|---|---|---|---|---|---|---|
| | | sr | sb | sr | sb | sr | sb | sr | sb |
| Zn | | - | - | - | - | - | - | - | - |
| Cr (6+) | 4.7 | - | <2.0 | <2.0 | 3.8 | <2.0 | <2.0 | <2.0 | <2.0 |
| Ni | | - | 25.5 | - | 125 | 99.5 | 43.4 | 76.7 | 51.6 |
| Cd | | - | - | - | - | - | - | - | - |

| Species | S2 | G5 | | G6 | | G7 | | G8 | |
|---|---|---|---|---|---|---|---|---|---|
| | | sr | sb | sr | sb | sr | sb | sr | sb |
| Zn | | - | - | <6.4 | <6.4 | 19.7 | <6.4 | <6.4 | 12 |
| Cr (6+) | 3.8 | 2.7 | <2.0 | <2.0 | <2.0 | <2.0 | <2.0 | - | 6.9 |
| Ni | | 43.5 | - | 120 | - | 61 | 56.3 | - | 59.4 |
| Cd | | - | - | - | - | - | - | <1.2 | <1.2 |

Note: "-" was not detected.

### 3.5. Preliminary Exploration of Solidification/Stabilization Scheme

Montmorillonite (MMT), lime (LM), zero valent iron (ZVI)) and sodium carboxymethyl-cellulose (CMC) were selected to mix each other in different proportions to stabilize heavy metals in this study. Two sediment samples (G2-sb and G6-sr) that the leaching concentration exceeded the threshold was studied, and the results are exhibited in Table 3. It can be seen that amendments formula for ZVI + CMC (9.5% + 0.5%) effectively reduced the leaching concentration of Ni in both sediment samples. The leaching concentration of Ni in sites G2-sb and G6-sr decreased by 57.04% and 93.16%, respectively, after treatment, far below the threshold (100 µg/L) of *SGQ*. CMC and its derivatives can be coordinated with polyvalent metal ions to form three-dimensional cross-linked structures with a retention effect [53]. Wu et al. [54] found that the complexes of CMC with Ni (II) were synthesized by the coordination reaction that the organic functional groups mainly are carboxymethyl and hydroxyl groups. The organic metal compound shows an octahedral geometry around the nickel, and the probable formula is $NiM \cdot 4H_2O$. IR also has been evidenced to remove Ni ions from water quickly and efficiently; the mechanism mainly included displacement, adsorption, complexation and other possible or several actions [55]. Similarly, Franco et al. [56] used ZVIcol (colloidal zerovalent iron) that synthesized with CMC and an ultrasound to remedy the fractions of Cr VI) (labile, exchangeable and insoluble) in soil, more than 98% of these species were reduced.

**Table 3.** Leaching content of heavy metals after adding stabilized reagents.

| No. | Regents | Adding Proportions | G2-sb Ni (µg/L) | G6-sr Ni (µg/L) |
|---|---|---|---|---|
| 1 | MMT + LM | 2% + 8% | $1.19 \times 10^3$ | $1.59 \times 10^3$ |
| 2 | MMT + LM | 4% + 6% | 530 | $1.05 \times 10^3$ |
| 3 | MMT + LM | 8% + 2% | 68.5 | 437 |
| 4 | MMT + ZVI | 2% + 8% | 50.1 | 331 |
| 5 | MMT + ZVI | 4% + 6% | 57.0 | 484 |
| 6 | MMT + ZVI | 8% + 2% | 158 | 705 |
| 7 | MMT + CMC | 9.5% + 0.5% | 221 | $1.56 \times 10^3$ |
| 8 | LM + ZVI | 2% + 8% | 23.3 | 192 |
| 9 | LM + ZVI | 4% + 6% | 51.5 | 135 |
| 10 | LM + ZVI | 8% + 2% | 404 | $1.96 \times 10^3$ |
| 11 | LM + CMC | 9.5% + 0.5% | 315 | $1.94 \times 10^3$ |
| 12 | ZVI + CMC | 9.5% + 0.5% | 53.7 | 8.2 |

Note: Adding proportions were the wet base ratio (w.w.). The moisture content of the sediment was about 65%.

Cement treatment is an effective method for enhancing the engineering behaviors of sediments and encapsulating contaminants [15]. Sulphate aluminum cement (SAC), CMC, ZVI and monopotassium phosphate (KP) were chosen to solidify/stabilize heavy metals in this study, and the results are shown in Table 4. The leaching concentration of Ni in site G2-sb and G6-sr decreased by 42.2% and 46.6%, respectively, after adding SAC + CMC (1% + 0.3%), and decreased by 73.3% and 34.1%, respectively, after adding SAC + ZVI + CMC (1% + 0.5% + 0.3%). These two amendments formulas could effectively draw down the leaching concentration of Ni, making it under the limit value (100 µg/L) of *SGQ*. The solidification of SAC was in no small part because of ettringite and aluminum glue generated in the early hydration stage, which solidified heavy metals mainly by means of ion replacement and chemical physical adsorption [57]. $Al^{3+}$ in ettringite was conformed to be substituted by $Cr^{3+}$, $Mn^{3+}$ and $Fe^{3+}$, and $Ca^{2+}$ can be replaced by $Cd^{2+}$, $Pb^{2+}$, $Zn^{2+}$, $Mn^{2+}$, $Ni^{2+}$ and $Fe^{2+}$ in ettringite [58]. Chen [59] found that the curing rates of ettringite for $Ni^{2+}$ could reach 87.9%. However, regent formulation incorporating KP did not appear to be effective for heavy metal fixation. Zhong et al. [60] discovered that the precursor of ettringite $(Ca\text{-}Al,\text{-}OH,\text{-}SO_4)$ could combine with phosphate to form stable phosphate ettringite, whether this process influenced its stability to heavy metals is unknown.

**Table 4.** Leaching content of heavy metals after adding cement curing reagents.

| No. | Reagents | Adding Proportions | G2-sb | G6-sr |
| --- | --- | --- | --- | --- |
|  |  |  | Ni (µg/L) | Ni (µg/L) |
| 1 | SAC + CMC | 1% + 0.3% | 72.3 | 64.1 |
| 2 | SAC + ZVI + CMC | 1% + 0.5% + 0.3% | 33.4 | 79.1 |
| 3 | SAC + KP + CMC | 1% + 0.5% + 0.3% | 239 | 758 |

Note: Added proportions were the dry base ratio (d.w.). The moisture content of the sediment was about 65%.

## 4. Conclusions

The heavy metal levels in the public river sediment were apparently higher than the background value. The distribution of metals was not strongly affected by the pH and OM contents, whereas they were associated with each other and may have common anthropogenic and natural sources. $I_{geo}$ of Hg showed moderately contaminated grade and other metals exhibited a none–moderately polluted degree. In comparison with the relevant national standards of China, the concentrations for all metals were below the threshold of *SEQD*, but the concentrations of Zn, Cr, Ni and Cd exceeded the limits of *PSG* and *SEQA*. The leaching experiment showed the concentration of Ni overtopped the threshold of *SGQ*. Thus, the dredged sediment could be reused for development land after solidification/stabilization. Three amendments formulations were ZVI and CMC, SAC and CMC, as well as SAC, ZVI and CMC, which effectively reduced leaching concentrations of heavy metals.

**Supplementary Materials:** The following supporting information can be downloaded at: https://www.mdpi.com/article/10.3390/w14142247/s1, Table S1: The information on method assurance; Table S2: Soil background values ($B_n$) in Jiangsu Province (mg/kg); Table S3: Pollution levels associated with the geoaccumulation index ($I_{geo}$) of the metals in sediment; Table S4: Critical limits (mg/kg) for heavy metals in soils in some foreign countries; Table S5: Ecological Soil Screening Levels (Eco-SSLs, mg/kg dry weight) for heavy metals in soil from USEPA; Table S6: Screening values (mg/kg) for heavy metals in soil in the national standard of China; Table S7: Screening values (µg/L) for heavy metals in water in the national standard of China.

**Author Contributions:** Conceptualization, Q.X. and F.Z.; methodology, Q.X. and F.Z.; validation, B.W., X.C. and L.Y.; formal analysis, Q.X. and F.Z.; investigation, Q.X. and F.Z.; resources, L.Y., X.F., J.C. and T.Y.; data curation, Q.X. and F.Z.; writing—original draft preparation, Q.X.; writing—review and editing, F.Z., B.W., X.F., X.C. and L.Y.; visualization, Q.X., F.Z. and X.C.; supervision, L.Y., X.F., J.C. and T.Y.; project administration, L.Y., J.C. and T.Y. and funding acquisition, L.Y., J.C. and T.Y. All authors have read and agreed to the published version of the manuscript.

**Funding:** This article is funded by Macau University of Science and Technology Faculty Research Grants (project number: FRG-22-051-MSB).

**Data Availability Statement:** The data presented in this study are available in the article.

**Acknowledgments:** We would like to express our sincere gratitude to the State Key Laboratory of Pollution Control and Resource Reuse for providing equipment and advice during this study.

**Conflicts of Interest:** The authors declare no conflict of interest.

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
