# Peer review of "Assessment of Heavy Metal Pollution in Suburban River Sediment of Nantong (China) and Preliminary Exploration of Solidification/Stabilization Scheme"

_water, doi:10.3390/w14142247_

Round 1
Reviewer 1 Report
The article is topical as it address not only sediment pollution, but also their utilisation methods. The study is based on a significant volume of experimental work and the use methods seems to be relevant. The stabilisation methods are compared and the optimal solutions are suggested.
Some minor weakneses:
1. The conclusion part is too short and should be improved to reflect the content of the study and consequences of it
2, Some language problems can be found throughout the article. It should be improved, for example
znic (Zn)
Fig 2. the distribution of hea
y metals for Zn, Cr, Cu and Ni were all greater than the critical limits for foreign
Reviewer 2 Report
The document "Evaluation of heavy metal pollution in suburban river sediments of Nantong (China) and preliminary exploration of the solidification / stabilization scheme" aims to study the distribution of heavy metals along the public river, evaluate their contamination levels and test the best solution for the stabilization of inorganic elements that are present with potentially dangerous concentration levels from an environmental point of view in the study area.
The work is well structured and presented, and discussed.
I suggest some minor revision of the text to improve understanding that is not always clear. Check below for additional detailed comment:
- abstract lack of the main goals of the manuscript;
-pag.3 line 118-121: It is not clear if the sediment cores are sectioned longitudinally (i.e. each slice is divided in an half) or if the cores are divided into two section (one more superficial and one more deep), and if it is this the case, how thick are the sections ? Does the separation occur on the same thickness among cores?
-pag. 4 lines 168-169: there is a ripetition;
-pag.5 table 1: please insert the error of the measure in the table;
-pag.6 lines 209-210 and lines 215-217: sentencies are no clear;
pag.8 lines 244-245: the sentence is no clear;
Reviewer 3 Report
Abstract can end with some implications of the findings in a broader context. For example, what can others learn from your investigation? How can they apply your findings to their own case studies?
Lines 36-37: Please add some related references such as: Alarming carcinogenic and non-carcinogenic risk of heavy metals in Sabalan dam reservoir, Northwest of Iran.
Line 108: How did you conclude that “Thus, river water quality was rapidly deteriorating.”? This statement should be supported by related references.
The authors should place this work into a broader setting. The literature review does help in this regard. More importantly, Discussion section can be expanded to provide your perspectives on the implications of the finding to the broader community. Please compare your results with the similar works applied different indices to evaluate different types of water bodies (e.g., Metal contamination assessment in water column and surface sediments of a warm monomictic man-made lake: Sabalan Dam Reservoir, Iran)
Round 2
Reviewer 3 Report
The Authors have responded to my comments. My suggestion is acceptance.
Congratulations to the Authors
This manuscript is a resubmission of an earlier submission. The following is a list of the peer review reports and author responses from that submission.
Round 1
Reviewer 1 Report
Manuscript ID: water-1482260
Title: Assessment of heavy metal pollution in suburban river sediment of Nantong (China) and preliminary exploration of solidification/stabilization scheme
Authors: Qinqin Xu, Fengbin Zhao, Boran Wu, Xin Fang, Xuan Lin, Xiaoli Chai, Liqun Yuan *
This manuscript reported the concentrations of 8 heavy metals in sediments from two field campaigns, and then investigated the leaching and solidification of selected sediments. Personally, this paper has little scientific novelty, the writing needs extensive revision and improvements to meet the publication criteria. This manuscript is not recommended to be published in Water.
Major concerns:
- Extensive editing of English language and style is required. For example, a comma needed between Cr Pb (Line 19), a space needed before and after > and ± (Line 21: p>0.05; Line 156: 0.53±0.39), space needed between numbers and units (Line 81, Line 169, etc.). Pay attention to singular and plural (Lines 35 and 38: heavy metal). Rephrase your sentences to avoid plagiarism, such as Lines 31-32, 35, 41-43, 46-47, 104-106, 166-167, and many others. Correct typos (such as Line 153 Results and disccusion should be “discussion”) and grammar errors (such as Lines 175). Refine your language (Line 361: …reduce the leaching toxicity of Ni… this paper has no toxicity data).
- What is the scientific significance and novelty of this work? A simple survey of heavy metal concentrations in sampled sediments doesn’t warrant it being published in a peer-reviewed journal.
Minor concerns:
- The authors should give us a solid background, for example, why picked Montmorillonite (MMT), lime (LM), iron powder (IR), and sodium carboxymethyl-cellulose (CMC)? I think extensive studies have investigated the solidification/stabilization of heavy metals in contaminated soils. What do/don’t we know from previous studies? this is necessary to be stated in the Introduction.
- Line 135: why the pH of the leaching agent was so low 3.2? it is more like an acid-washing. Besides, the leaching experiment should be in a separate paragraph. Please add a description of the solidification experiment in Section 2.
- Line 155 and Line 170: eight kinds of heavy metals, 5 kinds of heavy metals, this is not an English writing style, please correct it.
- It is not necessary to list the published data and regulatory standards in tables (Tables 1, 3, and 4), they’re not your data.
- Line 172: for some cases, the concentration differed a lot (sr vs. sb). Please clearly and adequately interpret your results.
- Please clarify the rationality: As you mentioned in Lines 246-248, “sediment … is not suitable for green soil or agriculture soil, but it can be applied to public facilities land, public management/service land and commercial service facilities land.” But why compare sediment results with soil background, Planting Soil for Greening, Soil Environmental Quality: Risk Control Standard for Soil Contamination of Agricultural Land? Are there regulation standards specific for sediments or service/commercial lands? Likewise, it is not appropriate to compare the leaching data with the Standard for surface water quality.
Reviewer 2 Report
Review of “Assessment of heavy metal pollution in suburban river sediment of Nantong (China) and preliminary exploration of solidification/stabilization scheme” by Qinqin Xu , Fengbin Zhao , Boran Wu , Xin Fang , Xuan Lin , Xiaoli Chai , Liqun Yuan. The main objective is to investigate the distribution characteristics of heavy metals, assess pollution status of heavy metals, and explore an effective method for heavy metal stabilization. Please indicate what new research brings. The discussion should explain the difference between the results obtained and those obtained by previous researchers, trying to provide an answer to the possible differences. Are there concrete steps that can be recommended and how generalizable are the findings? The region in which the inquiry was conducted, What's distinctive about it? Author should add more results, it can be added in the attachment.
Reviewer 3 Report
Introduction is too brief (11 references) and do not much describe the state of art
As is not a metal
Materials and methods are poorly prepared and should be rewritten
- In Materials and methods the sampling device used should be described
- All analytical facilities which were used should be mentioned as well as qualifications of chemicals used
- Analytical methods – reference should be given
- The stabilization methods are even not mentioned: this a major problem!
- Statistics of the data treatment is not mentioned: statistical treatment of data is mandatory for high quality research!
In all Figures error bars should be indicated
In Results and Discussion the comparison with Chinese regulations should be expanded to address regulation also in other countries
Many formatting mistakes in the list of references
In Table 2 there is no need to indicate autocorrelation
The weakest part of the article is the analysis of leaching and stabilization as no experimental details are provided and no discussions about mechanisms behind the stabilization effects. Some results seems to be quite doubtful, for example stabilization with carboxymethylcellulose.